# Joint morphogenetic cells in the adult mammalian synovium

Anke J. Roelofs[1,*], Janja Zupan[1,*], Anna H.K. Riemen[1], Karolina Kania[1], Sharon Ansboro[1], Nathan White[1], Susan M. Clark[1] & Cosimo De Bari[1]

The stem cells that safeguard synovial joints in adulthood are undefined. Studies on mesenchymal stromal/stem cells (MSCs) have mainly focused on bone marrow. Here we show that lineage tracing of *Gdf5*-expressing joint interzone cells identifies in adult mouse synovium an MSC population largely negative for the skeletal stem cell markers *Nestin*-GFP, Leptin receptor and Gremlin1. Following cartilage injury, *Gdf5*-lineage cells underpin synovial hyperplasia through proliferation, are recruited to a *Nestin*-GFP[high] perivascular population, and contribute to cartilage repair. The transcriptional co-factor Yap is upregulated after injury, and its conditional ablation in *Gdf5*-lineage cells prevents synovial lining hyperplasia and decreases contribution of *Gdf5*-lineage cells to cartilage repair. Cultured *Gdf5*-lineage cells exhibit progenitor activity for stable chondrocytes and are able to self-organize three-dimensionally to form a synovial lining-like layer. Finally, human synovial MSCs transduced with *Bmp7* display morphogenetic properties by patterning a joint-like organ *in vivo*. Our findings further the understanding of the skeletal stem/progenitor cells in adult life.

[1] Arthritis & Regenerative Medicine Laboratory, Institute of Medical Sciences, University of Aberdeen, Aberdeen AB25 2ZD, UK. * These authors contributed equally to this work. Correspondence and requests for materials should be addressed to C.D.B. (email: c.debari@abdn.ac.uk).

The stem and progenitor cells that maintain and repair adult joint tissues, including articular cartilage, remain undefined. Cells that display mesenchymal stromal/stem cell (MSC) activity *in vitro* (that is, clonogenicity and multipotency) have been isolated from adult skeletal tissues including synovium[1,2] and bone marrow[3]. Recent efforts have focused on identifying skeletal stem cells in bone marrow *in situ* and understanding their contributions to bone homoeostasis and repair. In mouse bone marrow, perivascular MSCs are marked by Pdgfrα and Sca1 (refs 4,5), *Nestin* (*Nes*)-GFP[6,7] and Leptin receptor (LepR)[8–10], with at least partial overlap between these populations. They support haematopoiesis and are an important source of osteogenic and adipogenic lineages in adult marrow[6,10]. In addition, Worthley *et al.*[11] identified a population of osteochondroreticular (OCR) stem cells marked by expression of Gremlin (Grem)1, which are distinct from *Nestin*-Green Fluorescent Protein (*Nes*-GFP)$^+$ cells, and contribute to early postnatal growing bones and fracture repair in adult mice. In contrast to the significant advances in our understanding of the MSC populations in bone marrow, little is known about the MSCs that are resident in the synovial joint and their roles in the maintenance and repair of articular cartilage which, as opposed to the transient cartilage of the growth plate and fracture callus that undergo endochondral ossification, is stable throughout life.

The synovium is a thin membrane that lines the inside of synovial joints. It consists of a lining layer of macrophage-like synoviocytes and fibroblast-like synoviocytes (FLS), and a sub-lining vascularized connective tissue. Using a double-nucleoside-analogue labelling method, we showed the presence of label-retaining cells in synovium with an MSC phenotype[12]. The origins of these cells in synovium and their relationship with the MSC populations as described in bone marrow remain to be elucidated.

During embryonic life, condensation of mesenchymal cells leads to the formation of the cartilage template in the developing limb. In the region of a prospective synovial joint, a stripe of mesenchymal tissue forms the joint interzone (JI), which is marked by expression of a number of genes, including growth and differentiation factor 5 (*Gdf5*)[13]. In *Gdf5-Cre;LacZ* mice, Cre-induced expression of *LacZ* is detectable in nearly all JI cells of the prospective knee at E14.5 (ref. 14) and is seen in tissues of postnatal synovial joints[14,15]. Here we sought to determine the location, phenotype and function of *Gdf5*-expressing JI progeny in the adult knee, hypothesizing that they represent an ontogenetically defined population containing stem/progenitor cells that play important roles in joint homoeostasis and repair. Our findings show that *Gdf5*-lineage cells in mice persist in the adult synovium as MSCs with joint progenitor activity that after injury, in a Yes-associated protein (Yap)-dependent manner, proliferate to underpin synovial lining hyperplasia and contribute to cartilage repair. We further show that adult human synovial MSCs display joint morphogenetic properties *in vivo*. Together, our findings indicate that the synovium is a postnatal reservoir of MSCs that descend from the embryonic JI and maintain joint tissues in adult life.

## Results

### *Gdf5*-lineage cells persist as MSCs in adult synovium.
To trace cells of the embryonic JI, *Gdf5-Cre* mice were crossed with *tdTomato* (*Tom*) fluorescent reporter mice. In the *Gdf5-Cre* model, Cre-mediated recombination in the limbs is restricted to JI cells[14,15]. Mice with leaky, widespread Cre expression[14] were prospectively identified (Supplementary Fig. 1a–c). In adult knee joints, Tom$^+$ cells were present in articular cartilage, menisci, ligaments (Fig. 1a), epiphyses (Fig. 1b), synovium (Fig. 1c) and fat

pad (Supplementary Fig. 1d). They included osteocytes (Fig. 1b) and osteocalcin$^+$ osteoblasts (Supplementary Fig. 1e) in subchondral bone, but were rare in metaphyses or diaphyses (Fig. 1a and Supplementary Fig. 1f).

Tom$^+$ cells persisted in synovium up to at least 1 year of age (Fig. 1d). They were most abundant in synovial lining, but also included a small population of cells in the vascularized sub-lining connective tissue (Fig. 1e). Tom$^+$ cells expressed the pan-fibroblast/MSC marker Pdgfrα (Fig. 1f), and were distinct from CD16/CD32$^+$ macrophages (Fig. 1g) or CD31$^+$ endothelium (Fig. 1h). Lubricin (or Proteoglycan 4, Prg4) was expressed by both Tom$^+$ and Tom$^-$ cells in synovial lining, while undetectable in Tom$^+$ cells in sub-lining (Fig. 1i). Phenotyping by flow cytometry (Supplementary Fig. 2a–c) confirmed Tom$^+$ cells to be negative for CD45, CD11b, CD16/CD32 and CD31, while they heterogeneously expressed MSC/fibroblast markers, including Pdgfrα, Sca1, gp38, CD73, CD200, Thy1.1 (CD90), CD105 and alphaV (CD51) (Fig. 1j and Supplementary Fig. 2d). Similar results were obtained at ∼1 year of age (Supplementary Fig. 2e,f).

Sorting of freshly isolated cells for Tom and Pdgfrα expression within the CD45$^-$ stromal fraction (Supplementary Fig. 3a,b) showed Tom$^+$ and Tom$^-$Pdgfrα$^+$ populations to be enriched in colony-forming unit fibroblast (CFU-f) activity compared to Tom$^-$Pdgfrα$^-$ stromal cells (Fig. 1k and Supplementary Fig. 3c,d), and the latter failed to expand beyond p2.

Correlating *Tom*, *Gdf5* and *Cre* mRNA expression in embryonic limbs, adult knee soft tissues, and Tom$^+$ and Tom$^-$ sorted and culture-expanded cells (Supplementary Fig. 4) supported the notion that Tom expression in this model indicates derivation from the embryonic JI. Taken together, these findings show *Gdf5*-lineage cells to be major contributors to the clonogenic stromal cell populations in adult synovium.

### Relationship with previously identified skeletal stem cells.
The presence of Tom$^+$ cells near CD31$^+$ blood vessels in the vascularized synovial sub-lining (Fig. 1h) raised the possibility that the *Gdf5* lineage may include perivascular MSCs marked by *Nes*-GFP[6,7] or LepR expression[8–10]. In E14.5 *Gdf5-Cre;Tom;Nes-GFP* embryo hindlimbs, *Nes*-GFP$^+$ cells were found along the perichondrium (Fig. 1l) as reported[16]. They were flanking the JI region of the prospective knee where they co-localized with CD31 expression, but were distinct from Tom$^+$ JI cells (Fig. 1l,m). In adult synovium, a small population of *Nes*-GFP$^+$ cells was detected in the vascularized sub-lining tissue frequently associated with CD31$^+$ blood vessels that showed little overlap with Tom$^+$ cells (Figs 1m and 3a). Similarly, very few LepR-expressing cells were present within the *Gdf5*-lineage population (Fig. 1n). In bone marrow, an MSC population distinct from *Nes*-GFP$^+$ cells is identified by expression of Grem1 (ref. 11). Immunostaining revealed rare Grem1$^+$ cells located in the sub-lining synovial tissue, but no overlap was detected with *Gdf5*-lineage cells (Fig. 1o). These findings indicate that the *Gdf5* lineage in adult synovium is largely distinct from known skeletal stem/progenitor cell populations as described in bone marrow.

### *Gdf5*-lineage cells in synovium proliferate after injury.
Injury of the articular cartilage induces synovial hyperplasia, which we have shown to involve proliferation of MSCs[12,17]. We thus assessed the response of *Gdf5*-lineage cells in synovium to cartilage injury (Fig. 2a). Six days after injury, the hyperplastic synovium at the femoropatellar junction, near the cartilage defect, showed an increase in the percentage of cells that were Tom$^+$ (Fig. 2b,c; 61.7 ± 2.2% (mean ± s.d., $n = 4$) in injured versus 32.4 ± 5.8% (mean ± s.d., $n = 4$) in control). This was not due to

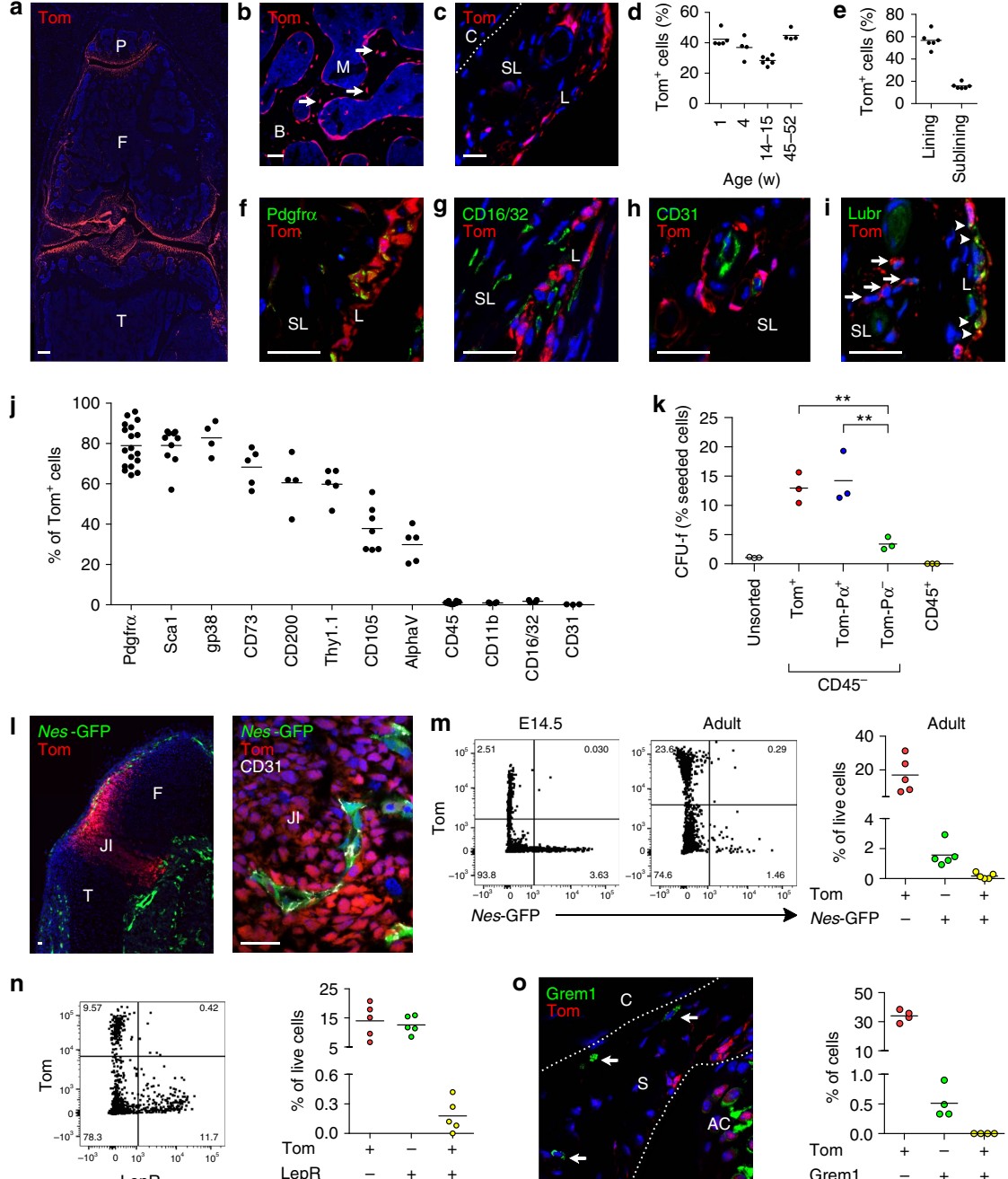

**Figure 1 | *Gdf5*-lineage cells in the mouse knee.** (**a–c**) Tom fluorescence (red) in adult *Gdf5-Cre;Tom* mice showing (**a**) low magnification overview of knee (*n* = 3), (**b**) epiphyseal bone with arrows indicating Tom$^+$ osteocytes (*n* = 3), (**c**) synovium (*n* = 6). Nuclei were counterstained with DAPI (blue). (**d,e**) Tom$^+$ cells in (**d**) synovium at different ages (*n* = 4-6), and (**e**) adult synovial lining and sub-lining (*n* = 6), as percentage of DAPI$^+$ cells. (**f–i**) Tom$^+$ cells (red) in synovium (**f**) express the MSC/fibroblast marker Pdgfrα (green; *n* = 4), are distinct from (**g**) CD16/CD32$^+$ (green) macrophages (*n* = 4) and (**h**) CD31$^+$ (green) endothelial cells (*n* = 3), and (**i**) express lubricin (Lubr; green) in lining (arrowheads) but not sub-lining (arrows) (*n* = 4). Nuclei were counterstained with DAPI (blue). (**j**) Cell surface phenotype determined by flow cytometry (*n* = 3-18; pooled data from seven experiments). See also Supplementary Fig. 2. (**k**) CFU-f activity of cells sorted by FACS. Colonies of ≥32 cells (≥5 population doublings) were counted (see also Supplementary Fig. 3). Tom$^+$ and Tom$^-$Pdgfrα$^+$ populations were enriched for CFU-f activity (**P < 0.01; *n* = 3; one-way ANOVA with Bonferroni post-test). (**l**) Tom (red) and GFP (green) fluorescence in hindlimb from *Gdf5-Cre;Tom;Nes-GFP* E14.5 embryo (*n* = 4), showing *Nes*-GFP$^+$ cells associated with CD31 staining (white) distinct from Tom$^+$ JI cells. Nuclei were counterstained with DAPI (blue). (**m**) Flow cytometry showing limited overlap between *Nes*-GFP$^+$ cells and Tom$^+$ *Gdf5* lineage in E14.5 hindlimb (*n* = 3) and adult knee synovium of *Gdf5-Cre;Tom;Nes-GFP* mice (*n* = 5). (**n**) LepR expression detected using a pan-LepR antibody showing very few LepR$^+$ cells within the Tom$^+$ *Gdf5* lineage (*n* = 5). (**o**) Grem1 (green) and Tom (red) expression in adult *Gdf5-Cre;Tom* mouse synovium (*n* = 4) detected by IF staining failing to show overlap. Nuclei were counterstained with DAPI (blue). P, patella; F, femur; T, tibia; S, synovium; L, synovial lining; SL, synovial sub-lining; C, capsule; B, bone; M, marrow; AC, articular cartilage; JI, joint interzone. Scale bars, 200 μm (**a**) and 20 μm (**b,c,f–i,l,o**).

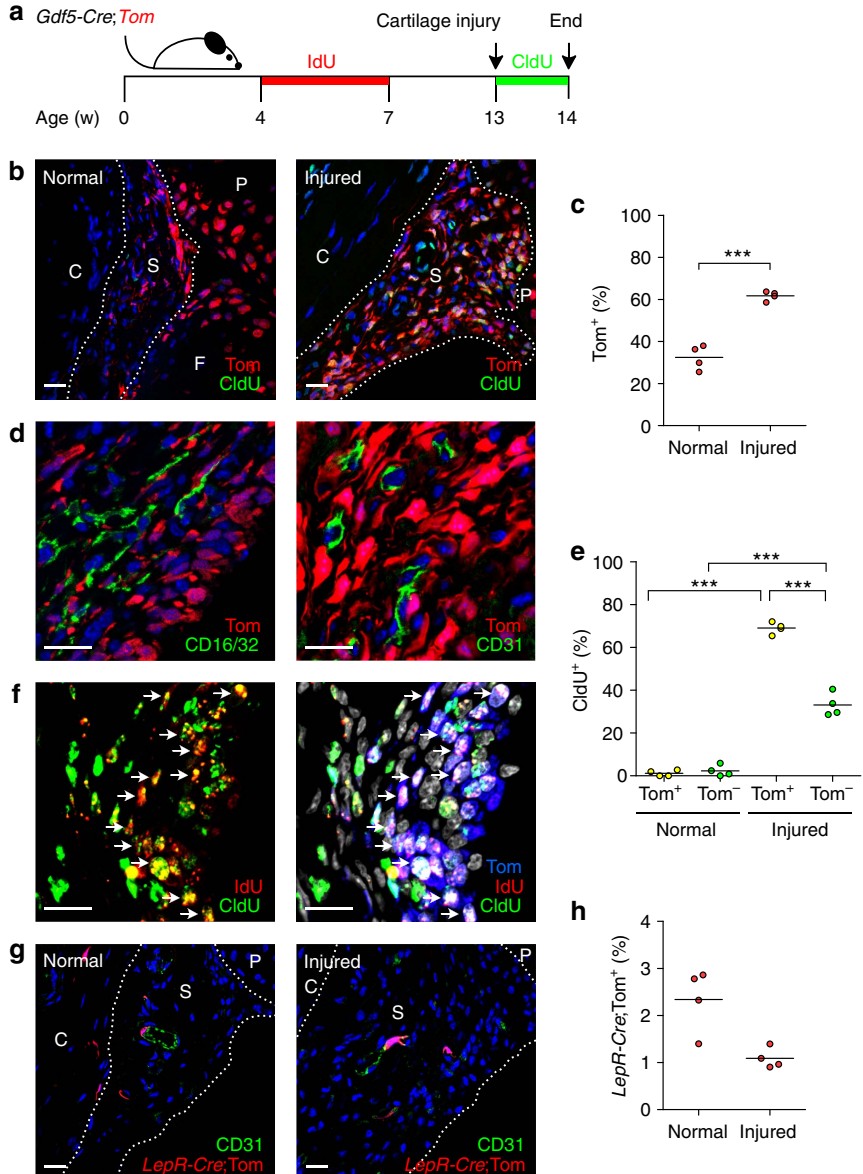

**Figure 2 | Contribution of *Gdf5* lineage to synovial hyperplasia after cartilage injury. (a)** Schematic experimental design for data in **b–f**.
**(b)** IF staining for Tom (red) and CldU (green) in control and injured knee synovium. Nuclei were counterstained with DAPI (blue). **(c)** Tom$^+$ cells, as percentage of total cells in synovium as shown in **b**, increased after injury (***$P < 0.0001$; $n = 4$; Student's $t$-test). **(d)** Tom$^+$ (red) cells in injured knee synovium negative for the haematopoietic marker CD16/CD32 (green; $n = 3$; image on left) and endothelial marker CD31 (green; $n = 4$; image on right). Nuclei were counterstained with DAPI (blue). **(e)** CldU-labelling showing higher rate of proliferation in Tom$^+$ compared to Tom$^-$ cells after injury (***$P < 0.0001$; $n = 4$; two-way ANOVA with Bonferroni post-test). **(f)** IF staining for IdU (red), CldU (green) and Tom (blue) in injured knee synovium ($n = 3$). Nuclei were counterstained with DAPI (grey). Left: IdU and CldU overlaid; right: all channels overlaid. Arrows indicate triple-labelled cells. **(g)** Tom$^+$ (red) cells in control and injured knee synovium of adult *LepR-Cre;Tom* mice ($n = 4$) with CD31 in green. Nuclei were counterstained with DAPI (blue). **(h)** Tom$^+$ cells in synovium as percentage of total cells. Dashed lines, synovial boundary; S, synovium; P, patella; F, femur; C, capsule. Scale bars, 20 μm. Data in **b,c,e** are representative of two experiments.

*de novo* induction of *Gdf5* or *Cre* expression (Supplementary Fig. 4), and phenotyping confirmed that Tom$^+$ cells remained distinct from CD16/CD32$^+$ haematopoietic and CD31$^+$ endothelial cells (Fig. 2d). Instead, the relative expansion of the Tom$^+$ population was likely the result of a high proliferative response to injury, as assessed by chlorodeoxyuridine (CldU) labelling (Fig. 2b,e; 69.0 ± 2.7% (mean ± s.d., $n = 4$) of Tom$^+$ versus 33.1 ± 5.4% (mean ± s.d., $n = 4$) of Tom$^-$ cells were CldU$^+$). Immunofluorescence (IF) staining for iododeoxyuridine (IdU) (marking label-retaining cells), CldU (marking cells proliferating after injury) and Tom revealed the presence of triple-positive cells in the hyperplastic synovium (Fig. 2f), indicating that the *Gdf5* lineage contained label-retaining (quiescent) cells that had re-entered the cell cycle in response to injury, a functional feature of stem cells postnatally[18].

To determine the contribution of perivascular MSCs to synovial hyperplasia, we analysed *LepR-Cre;Tom* mice. Similar to *Nes*-GFP$^+$ cells, the *LepR* lineage constituted a small population in the synovial sub-lining associated with CD31$^+$ blood vessels (Fig. 2g). In contrast to the *Gdf5* lineage, the *LepR*-traced population in synovium did not expand in response to injury (Fig. 2g,h).

Taken together, these data demonstrate that MSCs in synovium that are quiescent and proliferate in response to injury of the articular cartilage are contained for a large part within the *Gdf5* lineage.

**Recruitment into *Nes*-GFP$^+$ perivascular cells after injury.** Next, we used *Gdf5-Cre;Tom;Nes-GFP* mice to further investigate the relationship between perivascular cells and the *Gdf5* lineage in synovium following cartilage injury. The percentage of *Nes*-GFP single-positive cells in synovium did not change (Fig. 3a,b), suggesting a limited response of these cells to cartilage injury, similar to the *LepR* lineage (Fig. 2g,h). In contrast, the *Nes*-GFP$^+$Tom$^+$ population expanded (Fig. 3a,b; $0.9 \pm 0.1\%$ (mean ± s.d., $n = 4$) of total cells in injured versus $0.3 \pm 0.2\%$ (mean ± s.d., $n = 4$) in control). *Nes*-GFP$^+$Tom$^+$ cells were located in the sub-lining and were almost exclusively perivascular (Fig. 3a and Supplementary Fig. 5a). While the density of *Nes*-GFP single-positive cells surrounding blood vessels remained stable after injury, vascular density of *Nes*-GFP$^+$Tom$^+$ cells increased (Fig. 3c and Supplementary Fig. 5b,c; $4.4 \pm 1.1$ (mean ± s.d., $n = 4$) cells per mm vasculature in injured versus $1.8 \pm 1.1$ (mean ± s.d., $n = 4$) in control). Flow cytometry revealed *Nes*-GFP$^+$Tom$^+$ cells in injured knees to be mostly contained within the *Nes*-GFP$^{bright}$ fraction (Fig. 3d), and confirmed this population to be increased in response to cartilage injury relative to both Tom$^+$ (Fig. 3e; $5.9 \pm 3.8\%$ (mean ± s.d., $n = 12$) of Tom$^+$ cells were *Nes*-GFP$^{bright}$ in injured versus $0.9 \pm 1.4\%$ (mean ± s.d., $n = 12$) in control) and *Nes*-GFP$^{bright}$ populations (Fig. 3f; $25.8 \pm 13.4\%$ (mean ± s.d., $n = 12$) of *Nes*-GFP$^{bright}$ cells were Tom$^+$ in injured versus $9.1 \pm 12.1\%$ (mean ± s.d., $n = 12$) in control). Marker analyses showed that the *Nes*-GFP$^{bright}$Tom$^+$ cells were Pdgfrα$^-$Sca1$^-$CD105$^+$CD31$^-$LepR$^-$ (Fig. 3g,h), phenotypically similar to the *Nes*-GFP$^{bright}$ single-positive cells, but distinct from the bulk of the *Gdf5*-lineage population that were Pdgfrα$^+$Sca1$^+$, or the *Nes*-GFP$^{dim}$ population, which included subpopulations of Pdgfrα$^+$Sca1$^+$ cells, LepR$^+$ cells and CD31$^+$CD105$^+$ endothelial cells (Fig. 3g,h).

These findings were suggestive of a recruitment of *Gdf5*-lineage cells into a *Nes*-GFP$^{bright}$ population, described to localize around arterioles in bone marrow[19]. Staining of adjacent sections with either the pan-endothelial marker CD31, or SM22, a marker of vascular smooth muscle surrounding arterioles (Fig. 3i and Supplementary Fig. 5d), revealed a predominant localization of *Nes*-GFP$^+$Tom$^+$ cells with a *Nes*-GFP$^{bright}$ appearance tightly packed around arterioles. Interestingly, *Nes*-GFP$^+$Tom$^+$ cells themselves were frequently SM22$^+$, suggesting differentiation into smooth muscle cells.

Using an *in vitro* angiogenesis co-culture assay, we observed culture-expanded Tom$^+$ cells from *Gdf5-Cre;Tom;Nes-GFP* mice to give rise to *Nes*-GFP$^+$Tom$^+$ cells when co-cultured with sprouting mouse aortic rings (AR) (Fig. 3j), while no *Nes*-GFP-expressing cells were detected in wells without ARs.

Taken together, these findings show an injury-induced recruitment of *Gdf5*-lineage cells into a *Nes*-GFP$^{bright}$Pdgfrα$^-$ Sca1$^-$CD105$^+$CD31$^-$LepR$^-$ population that is predominantly periarteriolar and includes SM22$^+$ smooth muscle cells.

**Gdf5-lineage hyperplasia in synovium is Yap dependent.** We next sought to confirm a crucial role for *Gdf5*-lineage cells in synovial hyperplasia induced by cartilage damage, and to identify the molecular mechanism underpinning this process. Forced expression of a constitutively active form of the transcriptional co-factor Yap increased proliferation of MSCs *in vitro*[20]. Here we found Yap to be upregulated and prominently nuclear (active) in the hyperplastic synovium of injured mouse knees (Fig. 4a). For

clinical relevance, we analysed synovium from patients with intra-articular fracture or osteoarthritis. Irrespective of underlying pathology, nuclear (active) YAP was detected in areas of synovial lining with an activated, hyperplastic appearance, while YAP was barely detected in non-hyperplastic synovial lining within the same histological sections (Fig. 4b,c), or in normal knee joint synovium (Fig. 4d). YAP co-localized with the proliferation marker Ki67 (Fig. 4e) and CD55 (Fig. 4f), a marker of FLS in human synovial lining[21], while barely detectable in CD68$^+$ macrophages (Fig. 4f).

We therefore generated *Gdf5-Cre;Yap1$^{fl/fl}$;Tom* mice in which *Yap1* is knocked out in *Gdf5*-lineage cells marked by Tom expression (*Yap1* cKO mice), and compared them to *Gdf5-Cre;Yap1$^{fl/WT}$;Tom* haploinsufficient (*Yap1* cHa) mice and *Gdf5-Cre;Tom* controls. *Yap1* cKO mice were born at expected Mendelian frequencies, and were phenotypically normal with no obvious skeletal phenotype (Fig. 5a and Supplementary Fig. 6), indicating that *Yap1* expression by *Gdf5*-lineage cells is dispensable for joint development and homoeostasis. KO of *Yap1* in the *Gdf5* lineage was confirmed by IF staining of synovial cell cultures showing lack of Yap expression in Tom$^+$ cells (Fig. 5b). *Yap1* cKO did not affect the rate of colony formation *in vitro* (Fig. 5c); however, the proportion of large Tom$^+$ colonies was decreased, even from *Yap1* cHa mice, while the size of Tom$^-$ colonies was not affected (Fig. 5d).

Analysis of mice after cartilage injury revealed that the increased cellularity of synovial lining was prevented in *Yap1* cKO mice (Fig. 5e,f; $18.0 \pm 10.3$ (mean ± s.d., $n = 4$) average cells/section in *Yap1* cKO mice versus $85.9 \pm 28.1$ (mean ± s.d., $n = 6$) in controls). Strikingly, even *Yap1* cHa mice showed a decrease in cellularity of the lining compared to controls (Fig. 5e,f; $43.7 \pm 22.8$ (mean ± s.d., $n = 6$) average cells per section). In contrast, cellularity of the sub-lining, in which only a minority of cells are of *Gdf5* lineage (Fig. 1e), was not affected in either the *Yap1* cHa or *Yap1* cKO mice (Fig. 5e,g). Consistent with these findings, expansion of the Tom$^+$ *Gdf5* lineage in synovium following cartilage injury was diminished in *Yap1* cKO mice (Fig. 5e,h).

Altogether, these data point to Yap as a critical rate-limiting factor in the process of hyperplasia of the synovial lining that is driven by the *Gdf5* lineage.

**Gdf5-lineage cells give rise to new cartilage after injury.** We next focussed on the question whether adult *Gdf5*-lineage cells retain the ability, from embryonic development, to give rise to new cartilage. We injured *Gdf5-Cre;Tom* mice (which are on a mixed FVB and C57Bl/6 background) and analysed the cartilage repair tissue after 8–10 weeks (Fig. 6a). Four out of 12 injured knees exhibited a cartilage-like repair tissue that contained cells with chondrocyte morphology surrounded by a pericellular matrix staining for safranin O and collagen type II (Col2) (Fig. 6b), while the other knees contained a partial, fibrous-like repair tissue (Supplementary Fig. 7a). The majority of cells in the cartilage-like repair tissue were Tom$^+$ (Fig. 6b), though occasional clusters of Tom$^-$ chondrocytes were observed (Supplementary Fig. 7b). Knees with poor healing contained a lower percentage of Tom$^+$ cells in the repair tissue (Fig. 6c; $78.8 \pm 6.4\%$ (mean ± s.d., $n = 4$) in good versus $46.6 \pm 14.5\%$ (mean ± s.d., $n = 8$) in poor repair tissue). The cartilage-like repair tissue, but not the surrounding cartilage, contained Tom$^+$ chondrocytes that had incorporated and retained the nucleoside analogue CldU given for 12 days after injury (Fig. 6b), indicating cells had undergone transient proliferation following injury.

We also analysed seven areas of ectopic cartilage that had formed within the synovium not in direct continuum with the

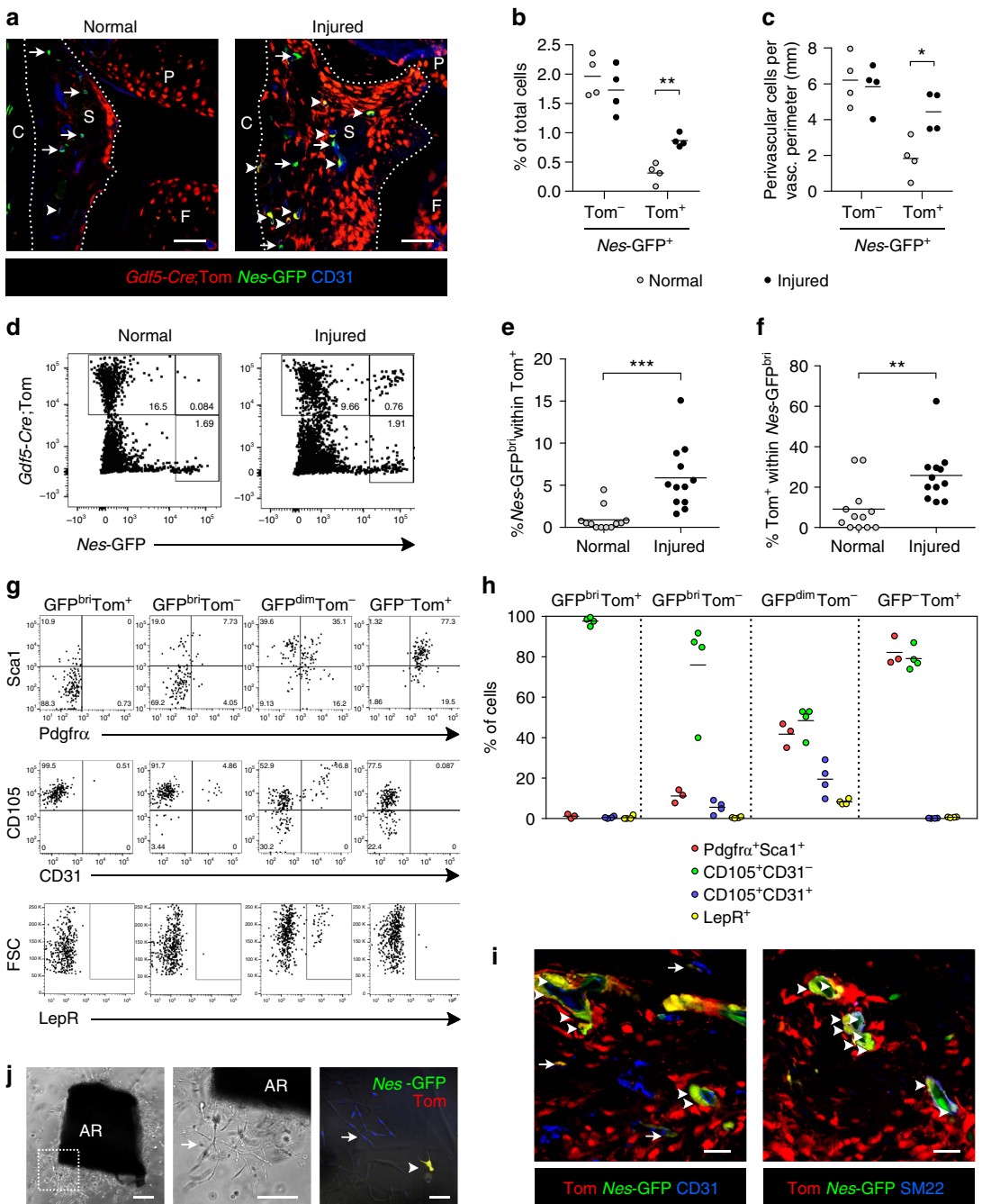

**Figure 3 | *Gdf5*-lineage recruitment into a *Nes*-GFP^bright perivascular population in response to injury.** (**a**) Tom^+ *Gdf5*-lineage (red) and *Nes*-GFP^+ cells (green) in control and injured knee synovium from adult *Gdf5-Cre;Tom;Nes-GFP* mice with CD31 in blue (*n* = 4). Arrows, *Nes*-GFP^+ cells; arrowheads, Tom^+ *Nes*-GFP^+ cells; dashed lines, synovial boundary; S, synovium; C, capsule; P, patella; F, femur. Scale bars, 50 μm. (**b**) Percentage of cells in synovium that were Tom^+ *Nes*-GFP^+ increased after injury (**P* = 0.0016; *n* = 4; Student's *t*-test). (**c**) Tom^+ *Nes*-GFP^+ perivascular cell number increased relative to the vasculature after injury (see also Supplementary Fig. 5a–c) (*P* = 0.0154; *n* = 4; Student's *t*-test). (**d**) Double-labelled cells in injured knee synovium were mostly *Nes*-GFP^bright. Equal total number of events is shown. (**e**,**f**) Tom^+ *Nes*-GFP^bright cells, quantified from **d**, increased after injury as percentage of (**e**) total Tom^+ (***P* = 0.0002; *n* = 12; Mann–Whitney test; pooled data from three experiments) and (**f**) total *Nes*-GFP^bright cells (**P* = 0.0060; *n* = 12; Mann–Whitney test; pooled data from three experiments). (**g**) Phenotypic analysis of labelled cells from injured knees showing similarity between Tom^+ *Nes*-GFP^bright and Tom^- *Nes*-GFP^bright populations. For each population, equal number of events is shown. FSC, forward scatter. (**h**) Percentage of cells within the indicated populations that were positive for the indicated marker(s) (*n* = 3–4; data from two experiments). (**i**) Adjacent histological sections stained with CD31 (blue; left) or SM22 (blue; right) showing Tom^+ (red) and *Nes*-GFP^+ (green) double-positive cells (appearing yellow) in injured knee synovium localizing predominantly around arterioles and expressing the smooth muscle marker SM22 (arrowheads) with some Tom^+ *Nes*-GFP^+ cells found around other CD31^+ vessels (arrows) (*n* = 4). Nuclei were counterstained with DAPI (blue). Scale bars, 20 μm. See also Supplementary Fig. 5d. (**j**) Mouse AR co-cultured with Tom^+ synovial cells from *Gdf5-Cre;Tom;Nes-GFP* mouse showing Tom^+ (red) and *Nes*-GFP^+ (green) double-positive cell (appearing yellow; arrowhead) in close proximity to endothelial tubules (arrow). Nuclei were counterstained with DAPI (blue). *n* = 8 replicate AR co-cultures. Left, middle: Phase-contrast only; box on left indicates region shown at higher magnification in middle. Scale bars, 200 μm. Right: Phase-contrast with fluorescence overlaid. Scale bar, 50 μm.

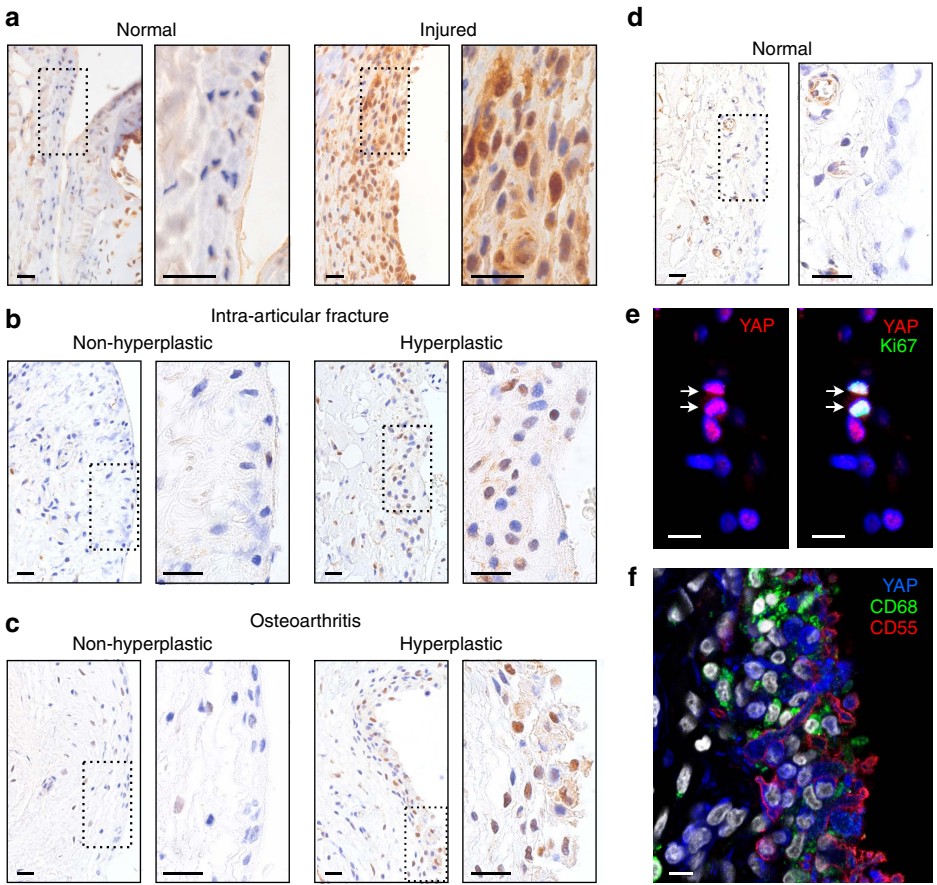

**Figure 4 | Yap is upregulated in the synovium of mice after injury and expressed in hyperplastic lining of human synovium.** (**a**) Yap (brown) expression in synovium of mice 4 days after cartilage injury ($n = 4$). Nuclei were counterstained with haematoxylin (blue). Boxed area on left is shown at higher magnification on right. Scale bars, 20 µm. (**b**–**f**) YAP expression in human synovium from (**b**) intra-articular fracture patients ($n = 3$ donors), (**c,e,f**) osteoarthritis patients ($n = 7$ donors) and (**d**) normal knee synovium ($n = 1$ donor). (**b**–**d**) IHC for YAP (brown) showing hyperplastic versus non-hyperplastic lining analysed within the same histological sections. Nuclei were counterstained with haematoxylin (blue). Boxed area on left is shown at higher magnification on right. Scale bars, 20 µm. (**e**) IF staining for YAP (red) and Ki67 (green). Nuclei were counterstained with DAPI (blue). Arrows indicate double-positive cells. Scale bars, 10 µm. (**f**) IF staining for YAP (blue) together with CD55 (red) marking synovial lining fibroblasts and CD68 (green) marking macrophages. Nuclei were counterstained with DAPI (grey). Scale bar, 10 µm.

joint surface[12]. Tom$^+$ cells with morphology of chondrocytes were found, to varying degrees, within all areas of ectopic cartilage (Fig. 6d and Supplementary Fig. 7c), and these included Tom$^+$ cells that had transiently proliferated following injury (Fig. 6d). The mosaic pattern of reporter expression in the newly formed cartilage-like tissues *in vivo* (Supplementary Fig. 7b,c), as well as the lack of reporter expression in chondrogenically induced Tom$^-$ sorted cells derived from *Gdf5-Cre;Tom* mice (Fig. 7c) suggests that Tom expression was not induced *de novo*. These findings demonstrate that the *Gdf5* lineage in adult synovium contains chondroprogenitors able to give rise to a cartilage-like tissue in the joint *in vivo*.

**Contribution to cartilage repair is Yap dependent.** Having shown that Yap is required in *Gdf5*-lineage cells for the synovial lining hyperplasia secondary to joint surface injury (Fig. 5), we analysed the repair tissue in *Yap1* cKO and *Yap1* cHa mice 8 weeks after cartilage injury (Fig. 6e). Strikingly, we detected fewer Tom$^+$ cells in the repair tissue in *Yap1* cKO mice compared to *Yap1* cHa controls (Fig. 6f; 14.9 ± 8.0% (mean ± s.d., $n = 4$) of total cells in *Yap1* cKO versus 53.1 ± 23.9% (mean ± s.d., $n = 3$) in *Yap1* cHa mice). Cartilage-like repair tissue consisting largely of Tom$^-$ cells was observed (Fig. 6g), consistent with the

occasional presence of Tom$^-$ chondrocyte-like cells in the repair tissue of *Gdf5-Cre;Tom* mice (Supplementary Fig. 7b). Notwithstanding, Tom$^+$ chondrocyte-like cells were detected in the repair tissue (Fig. 6h). Of note, results from *Yap1* cHa mice (Fig. 6f) were similar to those from *Gdf5-Cre;Tom* mice carrying wild-type *Yap1* described above (Fig. 6c). Taken together, these findings demonstrate that absence of Yap in *Gdf5*-lineage cells impairs their ability to populate the cartilage injury site.

***Gdf5*-lineage cells retain progenitor activity *in vitro*.** We then sought to investigate cell patterning and differentiation ability of culture-expanded cells (Fig. 7a). A synovial organoid culture model[22] was used to assess the ability of cells to undergo synoviogenesis. Tom$^+$ cells consistently self-organized into a lining layer of cells staining for the FLS marker cadherin-11 (Cdh11), reminiscent of the *in vivo* synovial lining, while Tom$^-$ cells lacked this ability (Fig. 7b).

Assessing chondrogenic ability showed Tom$^+$ cells to give rise to a cartilage-like matrix that more intensely stained for toluidine blue (TB) and Col2 compared with Tom$^-$ pellets (Fig. 7c). This prompted us to investigate the stability of the *in vitro*-induced cartilage-like phenotype using a subcutaneous implant model (Fig. 7a). After 4 weeks, 13 out of 17 Tom$^+$ (76%) and 2 out of 12

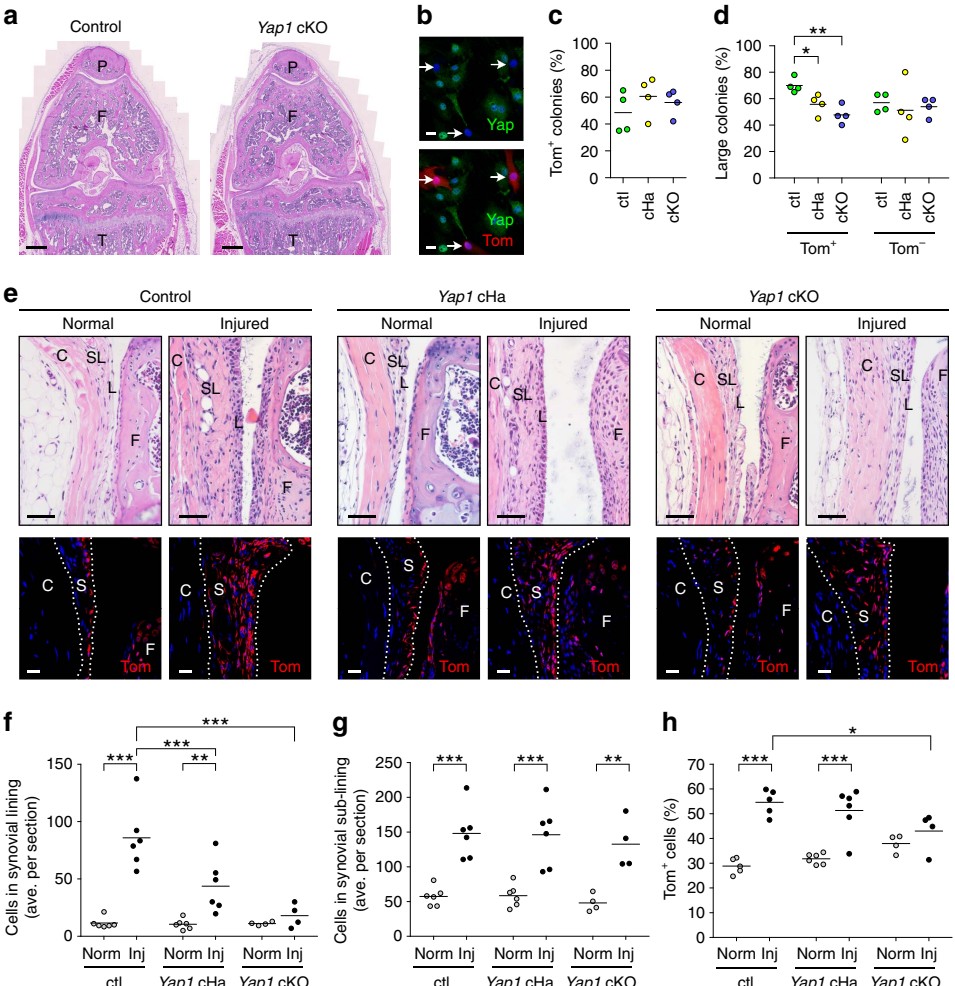

**Figure 5 | Conditional knockout of _Yap1_ in _Gdf5_-lineage cells prevents synovial lining hyperplasia after cartilage injury. (a**) H&E-stained sections of knees from an adult _Gdf5-Cre;Yap1^{fl/fl};Tom_ (_Yap1_ cKO) mouse and littermate control. P, patella; F, femur; T, tibia. Scale bars, 500 μm. (**b**) Lack of Yap (green) expression by Tom$^+$ (red) synovial cells from adult _Yap1_ cKO mice (arrows) indicating successful Cre-mediated _Yap1_ KO in _Gdf5_-lineage cells ($n = 3$). Nuclei were counterstained with DAPI (blue). Scale bars, 20 μm. (**c,d**) CFU-f activity of synovial cells isolated from control (ctl), _Yap1_ haploinsufficient _Gdf5-Cre;Yap1^{fl/WT};Tom_ (_Yap1_ cHa) and _Yap1_ cKO mice showing (**c**) percentage of colonies ($\geq 8$ cells, that is, $\geq 3$ population doublings) that were Tom$^+$, and (**d**) percentage of Tom$^+$ colonies that were large ($\geq 64$ cells, that is, $\geq 6$ population doublings) in ctl, _Yap1_ cHa and _Yap1_ cKO cultures (*$P < 0.05$; **$P < 0.01$; $n = 4$; one-way ANOVA with Bonferroni post-test). (**e**) Normal and injured knee synovium of ctl, _Yap1_ cHa and _Yap1_ cKO mice 6 days after cartilage injury. Top: H&E-stained sections. Bottom: Tom-stained (red) sections with DAPI (blue) counterstain. S, synovium; L, synovial lining; SL, synovial sub-lining; C, capsule; F, femur. Scale bars, 50 μm (H&E images) and 20 μm (fluorescent images). (**f,g**) Average number of cells per section in the synovial lining (**f**), and sub-lining (**g**), quantified from H&E images as in **e**, showing decreased cellularity in synovial lining but not sub-lining of injured cHa and cKO mice compared to ctl mice (**$P < 0.01$; ***$P < 0.001$; $n = 4-6$; two-way ANOVA with Bonferroni post-test). (**h**) _Yap1_ cKO mice showed diminished expansion of Tom$^+$ _Gdf5_-lineage cells in synovium after cartilage injury (*$P < 0.05$; ***$P < 0.001$; $n = 4-6$; two-way ANOVA with Bonferroni post-test).

Tom$^-$ pellets (17%) were recovered. The low retrieval rate of Tom$^-$ pellets suggested susceptibility to _in vivo_ resorption. Of the pellets analysed by histology, seven out of seven pellets (from three donors) had retained Col2 expression _in vivo_ with five out of seven pellets also showing TB metachromasia (Fig. 7d), while one out of two Tom$^-$ pellets retrieved showed Col2 and TB positivity. These data indicate the ability of adult _Gdf5_-lineage cells to give rise to a stable cartilage-like tissue _in vitro_.

We also transplanted Tom$^+$ cells orthotopically into cartilage defects in 8-week-old C57Bl/6 wild-type mice (Fig. 7a), which normally do not heal after cartilage injury[23]. After 4 weeks, Tom$^+$ cells were found in 7 out of 11 transplanted mice embedded in a repair tissue filling the original defect, which in five mice stained for Col2 (Fig. 7e), providing proof-of-concept for the ability of culture-expanded _Gdf5_-lineage cells to contribute to articular cartilage repair.

Tom$^+$ cells did not display osteogenic activity _in vitro_, while Tom$^-$ cells from the same mice within the same experiments were osteogenic to varying degrees (Fig. 7f). Assessment of three Tom$^+$ clonal populations derived by single-cell sorting showed all clonal populations, even after the extensive culture expansion required, to be synoviogenic, with one clonal population also displaying chondrogenesis and adipogenesis (Fig. 7g), pointing to a single-cell inherent multipotency within the _Gdf5_ lineage.

**Joint morphogenetic properties of adult human synovial MSCs.** Finally, we extended our investigation to MSCs from adult human synovium[1]. Culture-expanded cells from human synovium (but not dermal fibroblasts) display ability to undergo synoviogenesis _in vitro_[24], indicating their functional equivalence

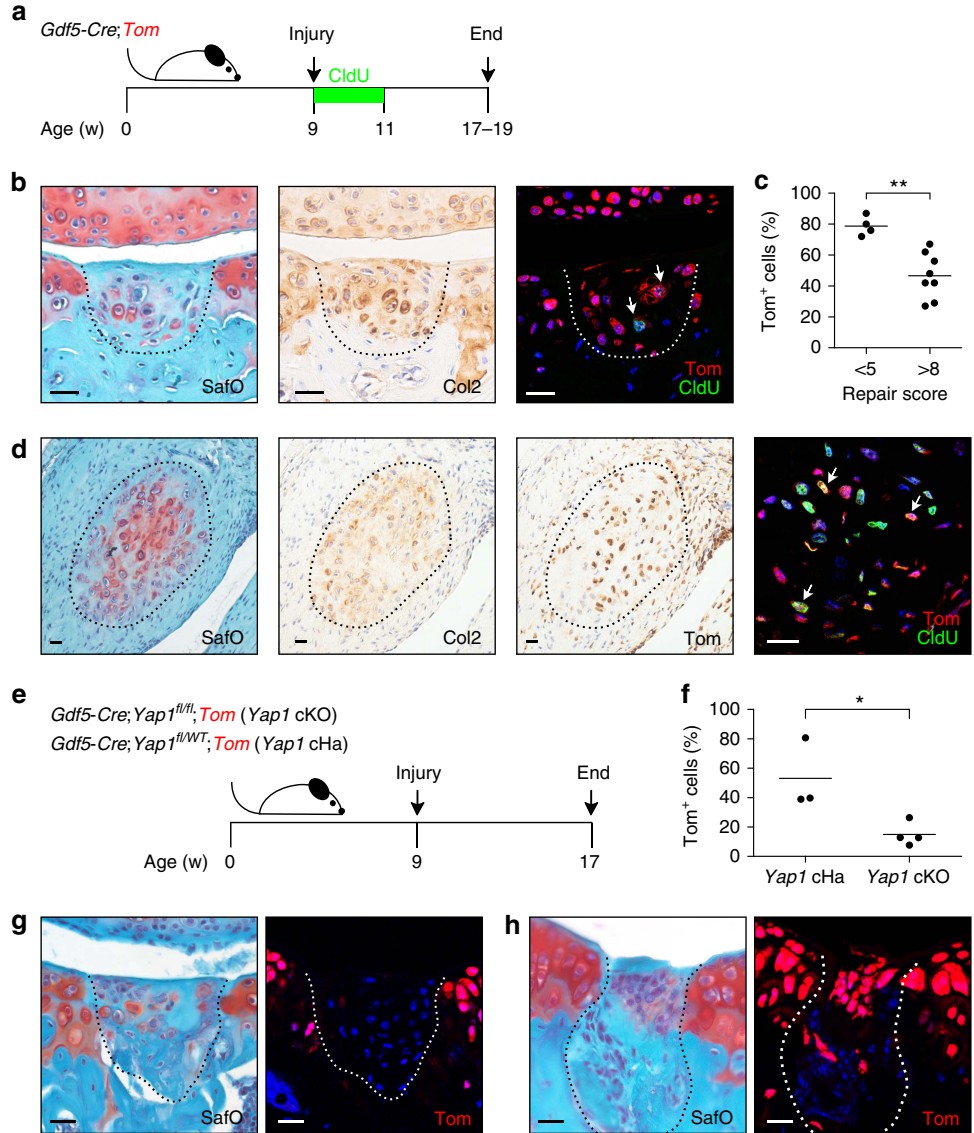

**Figure 6 | Contribution of *Gdf5*-lineage cells to cartilage formation and repair after joint surface injury.** (**a**) Schematic experimental design for data in (**b**–**d**). (**b**) Tom$^+$ cells in the articular cartilage repair tissue (indicated by dashed lines). Adjacent sections show (from left to right) safranin O/fast green (Saf O), Col2 (brown, with haematoxylin counterstain in blue), or Tom (red) and the proliferation marker CldU (green) with DAPI counterstain (blue). (**c**) Percentage of Tom$^+$ cells in the cartilage repair tissue in knee joints with low (that is, good, $n=4$) and high (that is, poor, $n=8$) repair scores (**$P=0.0020$; Student's $t$-test; pooled data from two experiments). (**d**) Tom$^+$ cells in ectopic cartilage-like tissue in synovium (indicated by dashed lines). Adjacent sections show (from left to right) Saf O, Col2 (brown, with haematoxylin counterstain in blue), Tom (brown, with haematoxylin counterstain in blue), or Tom (red) and CldU (green) with DAPI counterstain (blue). Arrows: Tom$^+$CldU$^+$ cells. Seven areas of ectopic cartilage in 4 mice were analysed. (**e**) Schematic experimental design for data in (**f**–**h**). (**f**) Fewer *Gdf5*-lineage cells were found in the repair tissue of *Yap1* cKO compared to *Yap1* cHa mice (*$P=0.0282$; $n=3$-4; Student's $t$-test). (**g**,**h**) Areas of joint surface repair largely devoid of Tom$^+$ cells (**g**), or containing Tom$^+$ chondrocyte-like cells (**h**) were both observed in *Yap1* cKO mice. Left: Saf O; right: Tom (red) with DAPI counterstain (blue). Scale bars in all panels: 20 μm.

to the mouse synovial *Gdf5*-lineage cells. Consistent with such a notion, we observed during culture expansion of unsorted synovial cells from *Gdf5-Cre;Tom* mice that Tom$^+$ *Gdf5*-lineage cells outgrew Tom$^-$ cells within several passages (Supplementary Fig. 8). Here we investigated the differentiation and patterning capability of human synovial MSCs during endochondral ossification induced by bone morphogenetic protein (BMP) signalling[25]. We adenovirally transduced culture-expanded human synovial MSCs to overexpress *Bmp7*, and 4 days later injected them into the posterior compartment of the thigh in a nude mouse model[26]. Surprisingly, after 4 weeks we retrieved an implant that histologically resembled a rudimentary joint, with

the two articulating sides each consisting symmetrically of articular cartilage, subchondral bone containing marrow and growth plate (Fig. 8a,b). Injection of untransduced cells resulted in no retrievable implant, while injection of human dermal fibroblasts transduced with Ad-*Bmp7* resulted in the formation of an ossicle consisting of a perimeter of cortical bone with trabeculae and a cavity containing fatty marrow, with residual cartilage remnants (Fig. 8c), similar to the previously reported ossicle formed after intramuscular implantation of Ad-BMP7 mixed with demineralized bovine bone powder[27]. *In situ* hybridization (ISH) for human-specific *ALU* genomic repeats showed the persistence of human cells in the joint-like

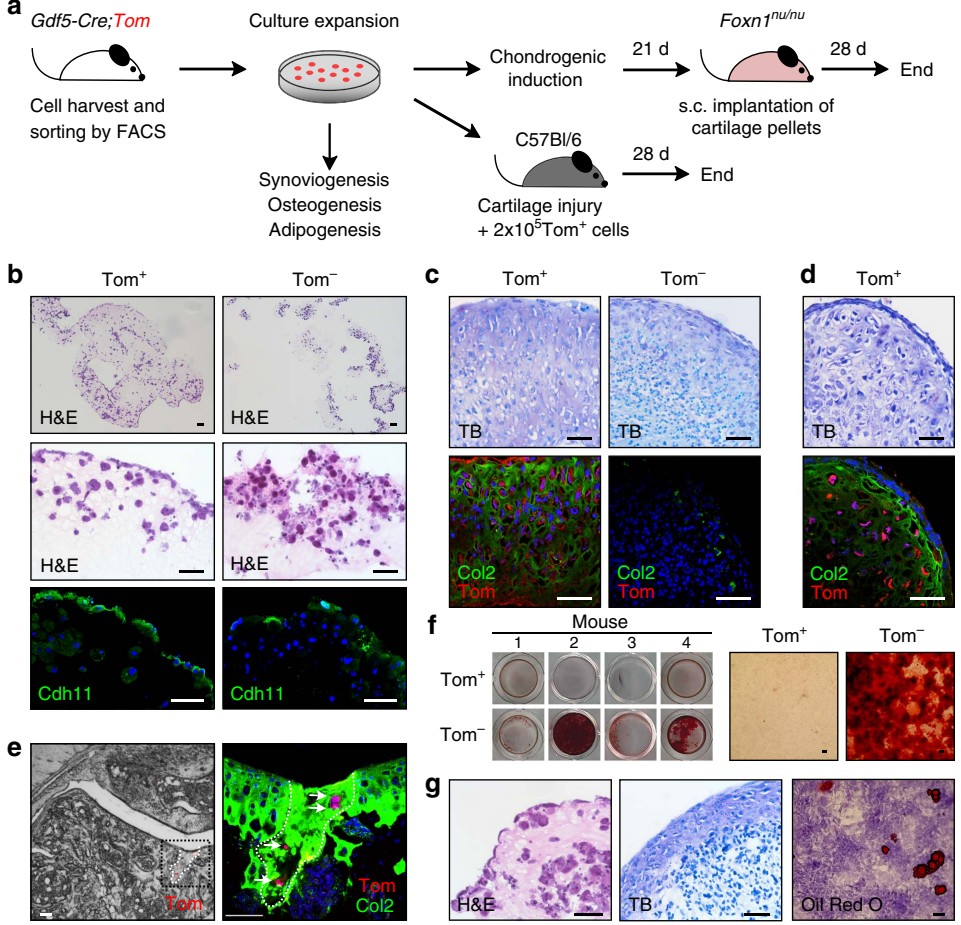

**Figure 7 | Cell patterning and differentiation capacity of culture-expanded *Gdf5*-lineage cells.** (**a**) Schematic experimental overview. (**b**) Tom$^+$ cells ($n = 4$ donor mice), but not Tom$^-$ cells ($n = 3$ donor mice), formed a synovial-like structure with elongated, flattened cells at the matrix-media interface staining for Cdh11. Top and middle: H&E-stained sections; bottom: IF staining for Cdh11 (green) with DAPI counterstain (blue). (**c,d**) Cartilage pellets of (**c**) Tom$^+$ and Tom$^-$ cells after *in vitro* chondrogenic induction (six pellets of cells from three donor mice) and (**d**) Tom$^+$ cells after explant from nude mice (seven pellets of cells from three donor mice; pooled data from two experiments). Top: TB staining. Bottom: Col2 (green) and Tom (red) IF, with DAPI counterstain in blue. Tom$^+$ cells gave rise to a matrix that stained for TB and Col2, resistant to *in vivo* remodelling. (**e**) Contribution of transplanted Tom$^+$ cells to cartilage repair in non-healer C57Bl/6 mice ($n = 11$). Left: Cryosection showing Tom$^+$ (red) cells at the cartilage defect. Right: Higher magnification of black-boxed area on left after immunostaining for Col2 (green) with DAPI counterstain (blue). Shown is an overlay of two confocal images at different *z* positions. Dashed line, outline of injury. (**f**) Tom$^+$ cells ($n = 4$ donor mice) failed to show osteogenesis by Alizarin Red S staining; microscopy images shown on right. Data are from four experiments. (**g**) Single-Tom$^+$-cell-derived clonal population showing tripotency to FLS (left), chondrocytes (middle) and adipocytes (right). Three clonal cell populations were analysed. Scale bars, 50 μm.

structure, with human nuclei detected in the articular cartilage and subchondral bone, but negligibly in the growth plate (Fig. 8d), while only sparse human nuclei were detected in cartilage remnants and bone of the implant obtained with dermal fibroblasts (Fig. 8e). RT–PCR analysis of the implants using species-specific primers revealed that while both implants were in large part host-derived, expression of human *ACAN* (aggrecan), *Prg4* (lubricin) and *BGLAP* (osteocalcin), but not *COL10A1* (collagen type X), was detectable in the joint-like structure obtained with human synovial MSCs (Fig. 8f and Supplementary Fig. 9), confirming the data obtained by ISH. These findings show that the combination of an adult synovial MSC population with one morphogen regulating endochondral ossification is sufficient, after ectopic implantation into skeletal muscle, to reproduce a joint-like structure, and provide proof-of-concept that synovial MSCs retain morphogenetic properties in adult life.

## Discussion

Our study provides evidence for the existence of an ontogenetically defined progenitor cell population purposely functional in the synovial joint and distinct from hitherto reported MSC populations, thereby adding another critical tile to the emerging mosaic of the diversity of the stem/progenitor cells that maintain and repair skeletal tissues in adult life.

We took advantage of a mouse strain with healing capacity to show that the majority of cells contributing to articular cartilage repair are of *Gdf5* lineage. We detected *Gdf5*-lineage cells in adult synovium, subchondral bone marrow and articular cartilage, and any of these tissues could harbour the cartilage-repairing cells. Articular cartilage is known to contain cells with chondroprogenitor activity[28–30]. In addition, full-thickness joint surface defects provide access for marrow cells to the injury. Notwithstanding this, the contribution of *Gdf5*-lineage cells to ectopic cartilage formation in synovium points to

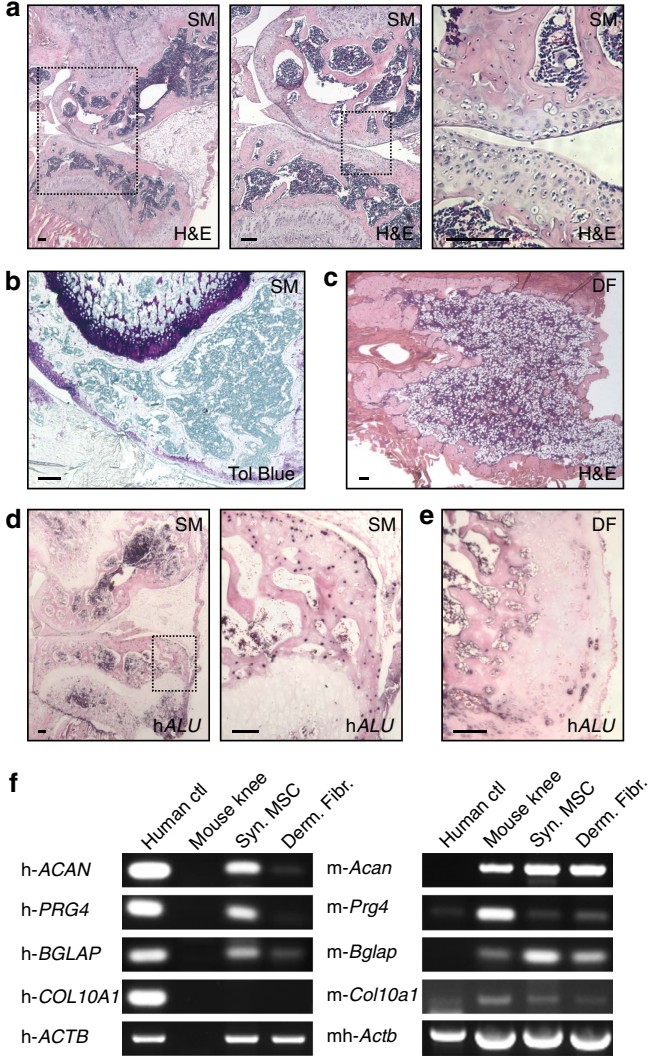

**Figure 8 | Joint morphogenetic properties of adult human synovial MSCs.** (**a,b,d,f**) Human synovial MSCs (*n* = 1) transduced with Ad-*Bmp7* and injected into nude mouse thigh formed a joint-like structure upon explant 4 weeks later, while (**c,e,f**) dermal fibroblasts (*n* = 1) transduced with Ad-*Bmp7* gave rise to an ossicle. (**d,e**) ISH for human-specific *ALU* genomic repeats. Black dots indicate human nuclei. Scale bars, 100 µm. Boxed areas on left shown at higher magnification on right. SM: synovial MSCs; DF: dermal fibroblasts; H&E: haematoxylin and eosin; Tol blue: toluidine blue; h*ALU*: human *ALU* repeats. (**f**) RT–PCR analysis using species-specific primers. Human positive controls (human ctl) were freshly isolated chondrocytes from articular cartilage for *ACAN* (aggrecan) and *PRG4* (lubricin), MSCs treated with vitamin D3 for *BGLAP* (osteocalcin), and bone marrow MSCs treated with TGFβ1 in pellet culture for *COL10A1* (collagen type X); a mouse knee joint was used as mouse ctl for all genes. h: primer pair specific for human; m: primer pair specific for mouse; mh: primer pair detects both mouse and human. Samples were equalized for h-*ACTB* (h-β-actin; human ctl used for h-*ACAN* and h-*PRG4* is shown), or for mh-*Actb* in the case of the mouse ctl. See also Supplementary Fig. 9.

chondroprogenitor activity in *Gdf5*-lineage cells in synovium. *Gdf5*-lineage cells were mainly present in the synovial lining where they included lubricin (Prg4)-expressing FLS. We also observed a population of *Gdf5*-lineage cells not staining for lubricin in the sub-lining tissue. Tracing of Prg4+ synovial lining cells suggested that most synoviocyte progenitors in 1-month-old mice do not express Prg4 (ref. 30). Altogether,

this indicates that the sub-lining tissue is a putative progenitor niche for FLS. In addition, following injury, *Gdf5*-lineage cells in the sub-lining were recruited into a periarteriolar *Nes*-GFP^bright population, which co-expressed CD105 and SM22 indicating a transition into myofibroblasts and smooth muscle cells[31], possibly to support vasculogenesis.

Similar to Grem1 OCR stem cells[11], *Gdf5*-lineage cells showed little overlap with perisinusoidal MSCs, and contribute both to skeletal development and to homoeostasis and repair. While we detected rare Grem1+ cells in adult synovium that showed no detectable overlap with the *Gdf5*-lineage population, the nature of the Grem1+ cells in synovium and whether these cells contain OCR stem cell activity similar to their counterparts in bone marrow, remain to be determined.

It is not known whether distinct stem cell populations in adult skeletal tissues co-exist. Chan *et al.*[32] proposed a model whereby skeletogenesis proceeds through a hierarchy of lineage-restricted progenitors, and one primitive skeletal stem cell gives rise to the cartilage, bone and stromal lineages, with differential expression of a combination of cell surface markers defining distinct stages of lineage commitment. While several of the key markers proposed (AlphaV, Thy1.1, CD105 and CD200)[32], and markers such as Pdgfrα and Sca1 which identify MSCs in mouse bone marrow[4,5], were expressed by *Gdf5*-lineage cells, we would interpret this as underscoring the dilemma of the tissue-dependent phenotypic identity of the MSCs and their poorly defined relationship with the fibroblasts, as for example Pdgfrα is considered a pan-fibroblast marker in skin[33].

Our findings indicate that Yap, a master regulator of gene transcriptional networks modulating stem cell activity and organ size[34], is dispensable in *Gdf5*-lineage cells for joint development and postnatal maintenance, but is required for their expansion underpinning synovial lining hyperplasia and their efficient population of the repair tissue following joint surface injury. The decreased contribution of *Gdf5*-lineage cells to cartilage repair in *Yap1* cKO mice may be due to an impaired recruitment to the injury site, as a role for Yap in cell migration has been reported[35,36], or an impaired cell proliferation, in keeping with our findings in the synovial lining. Of clinical relevance, Yap was expressed by FLS in hyperplastic synovial lining from patients with intra-articular fracture or osteoarthritis, although analysis of a larger number of patient samples will be needed for further validation and correlation with patient demographics and disease status.

In developing limbs, the *Gdf5*-expressing JI cells constitute a progenitor cell cohort endowed with joint formation and patterning capacity[37], with the spatiotemporal dynamics of *Gdf5* expression possibly instructing lineage divergence[38]. Our study shows that *Gdf5*-lineage cells in adult synovium retain joint progenitor activity. Moreover, we found that intramuscular injection of adult human synovial MSCs overexpressing *Bmp7* resulted in the formation of an ectopic joint-like structure, raising the fascinating prospect that these cells carry an imprinted code for joint morphogenesis. While BMP ligand specificity for inducing this morphogenetic process remains to be determined, it is intriguing to note that *Bmp7* during embryonic development is highly expressed in the perichondrium with characteristic interruption at the JI[39]. We speculate that cells that are progeny of the JI would define boundaries that steer the cascade of BMP-induced endochondral ossification into a joint morphogenetic process.

Our study highlights the importance of cell-lineage tracing experiments from defined developmental structures to identify, in adult tissues and organs, morphogenetic cell populations with the prospect of utilizing them for studies of organogenesis and regenerative therapies. Cell-based replacement therapies are

promising for degenerative conditions such as osteoarthritis, the commonest joint disease characterized by progressive loss of the articular cartilage. Our findings provide a scientific rationale to put forward cells from the adult synovium for joint surface regeneration and treatment of osteoarthritis.

## Methods

**Mice and _in vivo_ procedures.** _Gdf5-Cre_ mice (Tg(_Gdf5-cre-ALPP_)1Kng)[14] were donated by Dr David Kingsley (Stanford, CA, USA). _Nes-GFP_ mice (Tg(_Nes-EGFP_)33Enik)[40] were kindly provided by Dr Cristina Lo Celso (Imperial College London, UK), with permission from Dr Grigori Enikolopov (Cold Spring Harbor Laboratory, NY, USA). _LepR-Cre_ (_Lepr_tm3(cre)Mgmj) mice that express _Cre_ in cells expressing the long isoform of the leptin receptor (_LepRb_)[41] were kindly provided by Dr Lora Heisler (University of Aberdeen). _Yap1_fl mice (_Yap1_tm1.1Fcam)[42] were provided by Dr Fernando Camargo (Harvard, MA, USA). Cre-inducible tdTomato (Tom) (B6.Cg-_Gt(ROSA) 26Sor_tm14(CAG-tdTomato)Hze/J)[43] reporter mice were obtained from The Jackson Laboratory. Mouse breeding was undertaken at Charles River Laboratories, UK. _Gdf5-Cre_ mice were maintained on an FVB background, all other mice on a C57Bl/6 background. Genotyping was performed according to published protocols[14,40] or as recommended by The Jackson Laboratory or donating investigator. For analysis of embryos, mice were timed mated, and the morning of vaginal plug detection was designated day 0.5. At E14.5, pregnant dams were killed and embryos collected. C57Bl/6 wild type and immunodeficient BALB/c _Foxn1_nu/nu (nude) mice were purchased from Charles River or Harlan Laboratories. Animal experiments were performed at the University of Aberdeen's animal facility, where mice were kept in a temperature-controlled room on a 12:12 light-dark cycle and provided with _ad libitum_ food and water. Adult mice were analysed at 12–20 weeks of age. Both male and female mice were used for experiments. Cartilage repair was analysed in female mice only. _In vivo_ imaging was performed using a Bruker MS FX Pro imager. To detect Tom+ cells in the circulation, blood was withdrawn from the tail vein and Tom fluorescence detected on an LSRII flow cytometer (BD Biosciences, Oxford, UK).

Joint surface injury was performed by medial parapatellar arthrotomy as previously described[23]. Briefly, mice were anaesthetized and subjected to surgery under a dissection microscope. An incision was made to open up the skin over the knee joint area, followed by an incision along the medial side of the patellar ligament and through the quadriceps muscle to aid patellar dislocation. The patellar groove was exposed and a full cartilage thickness scratch along the length of the groove was made using a 25 g needle. The patella was then re-located, and the joint capsule/muscle and the skin were sutured separately. Quantifications of synovial cells were carried out in the lateral compartment of the knee at the femoropatellar junction. The contralateral leg served as internal uninjured control. Sham-operated controls were not included as we have found little response to sham surgery in the synovium of the lateral compartment of the knee in this model[12,17]. Proliferating cells were detected by administration of IdU and/or CldU as previously described[12]. For orthotopic cell transplantation, Tom+ sorted and culture-expanded cells from four _Gdf5-Cre;Tom_ mice were pooled and $2 \times 10^5$ cells in 1 μl PBS were applied to the cartilage defect using a Hamilton syringe (Hamilton, Bonaduz, Switzerland) before closing the joint.

Animal experiments were approved by the UK Home Office and the Animal Welfare and Ethical Review Committee of the University of Aberdeen or the KU Leuven Ethical Committee for Animal Research. Animal experiments were performed in compliance with the 3Rs.

**Human tissues.** Human synovial tissue samples were obtained after informed consent, with ethical approval from the NHS Grampian Biorepository. Synovial samples were obtained from seven patients with osteoarthritis (three males, four females; 50–81 years old) undergoing knee (six) or hip (one) joint arthroplasty, three patients with intra-articular fractures undergoing fracture fixation surgery at 2–8 days following injury (two males, one female; 20–80 years old; two glenohumeral joint fractures, one radial head fracture), and a patient (male adult) undergoing knee joint excision for malignancy at a site not involving the sampled area.

**Histology and immunostainings.** Samples for histology were fixed in 4% methanol-free paraformaldehyde (PFA; TAAB, Reading, UK) overnight, and decalcified in 4–10% EDTA (WVR Chemicals, Leuven, Belgium) for 2 weeks at 4 °C. For cryoembedding, samples were incubated in 15% sucrose (Sigma) for 1 h followed by 30% sucrose overnight at 4 °C, then snap-frozen in optimum cutting temperature (OCT) reagent (VWR Chemicals) over liquid nitrogen, and cut in 8 μm-thick sections using a cryostat (Leica Microsystems Ltd, Milton Keynes, UK). For paraffin embedding, samples were dehydrated, embedded in paraffin and cut into 5 μm-thick sections using a microtome (Leica Microsystems Ltd). Adult knee joints were embedded flexed as previously described[23]. Sections were rehydrated and stained with haematoxylin and eosin (H&E, WVR Chemicals), TB (HoverLabs, Haryana, India) or safranin O and fast green (Sigma), according to standard protocols.

Immunohistochemistry (IHC) and IF stainings were performed as previously described[12]. For cryosections, antigen retrieval was performed by incubation in 1% sodium dodecyl sulfate (Fisher Scientific) in PBS for 5 min. Primary antibodies used are listed in Supplementary Table 1. Sections were counterstained with haematoxylin for IHC, or 4′,6-diamidino-2-phenylindole (DAPI; Life technologies, Eugene, OR, USA) for IF. IHC sections were analysed using an Imager Axioskop 40 microscope (Carl Zeiss Ltd, Cambridge, UK) with ProgRes C14 camera and ProgRes CapturePro 2.8.8 software (JenOptik, Jena, Germany). IF sections were analysed on a Zeiss 700 or 710 META Laser-Scanning Confocal Microscope with ZEN software (Carl Zeiss Ltd), and transmitted light was captured to define the area of synovium. Some sections were tile-scanned on a Zeiss Axioscan Z1 slide scanner (Carl Zeiss Ltd). Image analysis and cell counting were performed using ZEN2010 (Carl Zeiss Ltd) and ImageJ software. For quantification of labelled cells in histological sections, 2–11 sections per knee joint were analysed, and the total number of counted cells determined.

**Phenotyping and sorting of mouse synovial cells.** Isolation of synovial cells from mouse knee joints was performed by digestion with 1 mg ml−1 collagenase type IV (Sigma) in normal growth media (high-glucose DMEM (Lonza, Verviers, Belgium) with 10% foetal bovine serum (FBS; ThermoFisher Scientific, Renfrew, UK), and 100 units ml−1 of penicillin and 0.1 mg ml−1 streptomycin (Sigma)) for 50 min at 37 °C under agitation, according to a protocol optimized for synovial cell isolation[44]. Cells were released by vigorous vortexing, and cell suspensions passed through a 40 μm cell strainer (Fisher Scientific, Loughborough, UK). Cells were seeded at 5,000 cells cm−2 and grown in Mouse Mesencult Proliferation kit (StemCell Technologies, Cambridge, UK) with penicillin/streptomycin and 2 mM L-glutamine (Sigma) until cultures were established, then cultured in regular growth media.

Freshly isolated cells were stained in 2 mM EDTA in PBS with 0.5% FBS in 96-well conical bottom plates with fixable viability dye eFluor780 (eBiosciences, Hatfield, UK) and antibodies listed in Supplementary Table 2. Data were acquired on an LSRII flow cytometer and analysed using FlowJo v10 software (Ashland, OR, USA). Unstained and single-labelled cells or antibody-labelled CompBeads (BD Biosciences) were used to set compensation and gates were applied based on fluorescence-minus-one controls. To identify single live cells within fresh cell isolates, erythrocytes and debris were gated out based on forward-side scatter profile, then doublets and aggregates were gated out based on forward scatter parameters, and finally dead cells were excluded based on viability dye staining (see Supplementary Fig. 2).

Sorting was performed on an Influx Cell Sorter (BD Biosciences). Erythrocytes and debris were excluded based on forward and side scatter profile, and cells were sorted based on Tom fluorescence, and in one experiment, CD45 and Pdgfrα IF staining. Small aliquots of sorted cells were analysed on an LSRII flow cytometer immediately after sorting to determine purity. For derivation of clonal cell populations, Tom+ cells were directly sorted into a 96-well plate. After 12 days of culture, eight wells were identified that contained a single colony, and three clones continued to expand and were used to assess potency at the single-cell level.

**CFU-f and potency assays.** For CFU-f activity, cells were seeded in 24-well plates in 4–8 replicate wells using Mouse Mesencult Proliferation kit with penicillin/ streptomycin and 2 mM L-glutamine at clonal seeding densities as established in pilot experiments. After 8 days, colonies were analysed on an Observer Z1 fluorescence microscope using AxioVision 4.8.20 software (Carl Zeiss Ltd).

Synovial organoid cultures (250,000 cells per organoid) were established as described[22], and after 21 days, fixed in 1% PFA, cryoembedded and sectioned for histological analysis. Chondrogenesis was induced by treatment with TGFβ1 (Gibco) in pellet culture (500,000 cells per pellet) for 21 days in serum-free DMEM with ITS-G (Gibco) supplemented with BSA (Sigma) and linoleic acid (Sigma), 0.1 μM dexamethasone and 50 μg ml−1 2-phospho-L-ascorbic acid trisodium salt (Sigma). To assess stability, chondrogenic cell pellets were implanted in subcutaneous pouches (up to four per mouse) of 11-week-old female nude mice, and after 4 weeks retrieved post-mortem under a stereomicroscope[45]. Pellets were embedded in paraffin and evaluated histologically and by IF staining. Pellets from cells that failed to show _in vitro_ chondrogenesis were excluded from analysis post-implant. For osteogenic differentiation, cells seeded at $4–5 \times 10^4$ cells cm−2 in 24- or 48-well plates were treated for 21 days with osteogenic media consisting of either α-MEM (Sigma) with 10% FBS and penicillin/streptomycin supplemented with 5 mM β-glycerophosphate disodium salt hydrate (Sigma) and 50 μg ml−1 2-phospho-L-ascorbic acid trisodium salt (Sigma), or MesenCult mouse osteogenic stimulatory kit (StemCell Technologies). Osteogenesis was detected by staining with Alizarin red S (Sigma). For adipogenic differentiation, cells seeded at $5 \times 10^4$ cells cm−2 in 48-well plates were treated for 21 days with adipogenic media consisting of Mesencult media supplemented with 500 μM dexamethasone, 10 μg ml−1 insulin (Sigma) and 10 μM indomethacin (Sigma). Adipocytes were detected by staining with Oil red O and cresyl violet (Sigma). Control wells received media without supplements.

Mouse AR angiogenesis assay was performed as described[46]. Briefly, ARs ∼0.5 mm in width from an 8-week-old C57Bl/6 mouse were incubated overnight in Opti-MEM with GlutaMAX-I (Gibco) with penicillin/streptomycin for serum starvation. Next day, ARs were embedded in growth factor-reduced Matrigel

(Corning) mixed with 3,000 culture-expanded *Gdf5-Cre;Tom;Nes-GFP* mouse knee joint cells and cultured in Opti-MEM with penicillin/streptomycin supplemented with 5% FBS and 30 ng ml$^{-1}$ VEGF (Peprotech) for 7 days, changing the medium every 2–3 days. Samples were fixed in 1% PFA, counterstained with DAPI and analysed on a Zeiss 710 META laser-scanning confocal microscope.

**Ectopic joint morphogenesis.** Adult human synovial MSCs[1] at p9 or human dermal fibroblasts[1] at p10 were transduced with pACCMVpLpA adenovirus encoding mouse *Bmp7* at 20 multiplicity of infection (MOI) and 4 days later $5 \times 10^6$ transduced cells were injected into the quadriceps muscle of a nude mouse[26]. Implants were analysed after 4 weeks histologically, by RT–PCR and ISH for human *ALU* genomic repeats.

**Gene expression analysis.** RNA was extracted from cells or tissues using either a standard TRIzol/chloroform extraction protocol, or a ReliaPrep RNA tissue Miniprep kit (Promega, Southampton, UK). Total RNA was reverse transcribed to cDNA. Expression of genes was determined by quantitative PCR (qPCR) using SYBR Green Master (Roche, Basel, Switzerland) on a LightCycler480 (Roche), as previously described[20], and data were normalized to expression of *Gapdh*. Detection of species-specific gene expression was carried out by semi-quantitative RT–PCR. Primer sequences are listed in Supplementary Table 3.

**ISH for human *ALU* genomic repeats.** ISH for human *ALU* genomic repeats was performed as described elsewhere[26]. Briefly, sections were deparaffinized and rehydrated. Matrix digestion was obtained by enzymatic treatment with 10 µg ml$^{-1}$ proteinase K in 0.1 M Tris-HCl, 50 mM EDTA, pH 8, for 30 min at 37 °C. Sections were immediately post fixed in 3% formaldehyde for 10 min, washed in PBS, dehydrated in an ethanol series and dried for 20 min at room temperature. After rehydration and equilibration in PBS, sections were acetylated in 0.25% acetic acid containing 0.1 M triethanolamine (pH 8) for 10 min, and pre-hybridized with 50% deionized formamide containing $4 \times$ SSC at 37 °C for 15 min. Sections were covered with hybridization buffer ($1 \times$ Denhardt's solution, 0.2 mg ml$^{-1}$ denatured sheared salmon sperm DNA, $4 \times$ SSC and 50% deionized formamide) containing 1 ng µl$^{-1}$ digoxigenin-labelled double-stranded DNA probe specific for human *ALU* genomic repeats and covered with a glass coverslip. Denaturation of both the probe and the genomic DNA template was achieved by heating the slides at 95 °C for 45 s. Hybridization was performed at 42 °C overnight. Sections were washed twice for 30 min at room temperature with $2 \times$ SSC and $0.1 \times$ SSC and for 30 min at 50 °C in $0.1 \times$ SSC. Digoxigenin was detected using a commercially available kit (DIG Nucleic Acid Detection Kit, Sigma) according to the manufacturer's protocol.

**Statistical analysis.** Statistical analysis was performed using GraphPad Prism v5 software. Data in graphs are shown as data points of individual mice, with lines indicating mean. Data in text are given as mean ± s.d. The *n*-number refers to number of mice unless otherwise stated. Mice that showed an unusual degree of transgene expression were excluded mostly *a priori* (Supplementary Fig. 1a–c) and occasionally at the analysis stage. Group assignment was based on genotype, therefore, no randomization was carried out. Sample size was estimated based on previous data where appropriate[12,23]. The researchers involved in the study were not completely blinded during sample collection or data analysis. For comparison of two groups, data were analysed by two-tailed unpaired Student's *t*-test, with or without Welch's correction for unequal variances, or by two-tailed Mann–Whitney test. One-way ANOVA with Bonferroni post-test was used for comparison of $\geq 3$ groups, or two-way ANOVA with Bonferroni post-test for comparisons across two variables. A *P*-value $\leq 0.05$ was considered statistically significant.

**Data availability.** All data supporting the findings of this study are available within the article and its Supplementary Information files, or are available from the corresponding author on reasonable request.

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

## Acknowledgements

The authors thank all members of the Arthritis & Regenerative Medicine Laboratory, particularly Dr Ana Sergijenko; Drs David Kingsley, Grigori Enikolopov, Fernando Camargo and Lora Heisler for sharing transgenic mice; Drs Henning Wackerhage, Neil Vargesson, Lynda Erskine, Chris Buckley, Francesco Dell'Accio and Frank Luyten for support and helpful discussions; Staff at the University of Aberdeen's Animal Facility, Microscopy & Histology Facility and Iain Fraser Cytometry Centre for their support. C.D.B. is grateful to Dr Frank Luyten's support for the experiment in Fig. 8, performed in his laboratory at KU Leuven, Belgium. We are grateful for the following funding: Arthritis Research UK (Grants No. 20050, 19429 and 20775), Medical Research Council (Grant No. MR/L020211/1) and Tenovus Scotland (Grant No. G13/14). A.H.K.R. is supported by the Wellcome Trust through the Scottish Translational Medicine and Therapeutics Initiative (Grant No. WT 085664).

## Author contributions

Conceptualization: A.J.R. and C.D.B.; Methodology: A.J.R., J.Z., K.K., S.M.C. and C.D.B.; Resources: A.J.R., A.H.K.R. and C.D.B.; Investigation: A.J.R., J.Z., A.H.K.R., K.K., S.A., N.W., S.M.C. and C.D.B.; Visualization: A.J.R., J.Z., A.H.K.R., K.K., S.A., N.W., S.M.C. and C.D.B.; Writing – original draft: A.J.R. and C.D.B.; Writing – review and editing: A.J.R., J.Z. and C.D.B.; Supervision: A.J.R. and C.D.B.; Funding acquisition: A.J.R., A.H.K.R. and C.D.B.

## Additional information

**Competing interests:** The authors declare no competing financial interests.

