## [Peer Review File · Nature Communications]

Reviewers' comments:

Reviewer #1 (Remarks to the Author):

Prior work has indicated that GDF5-Cre expressing cells in the joint can give rise to all tissues in this structure including the synovial cells, the ligaments and the articular cartilage. In the present study, the authors analyze both the proliferative and chondrogenic potential for synovial cells marked by GDF5-Cre expression. I think that this is a very carefully performed fate mapping study, which makes several interesting and novel observations: In particular, I thought that the finding that GDF5-Cre expressing synoviocytes proliferate following joint injury in a YAP-dependent fashion to be quite interesting. In addition, the authors demonstrate that GDF5-Cre expressing synoviocytes are able to reconstitute articular cartilage following joint injury in vivo, or following expansion of such cells in vitro and implantation of such cells into a joint injury model in vivo. Lastly, the authors present very striking data suggesting that Human synovial MSCs transduced with a Bmp7-encoding adenovirus can form a rudimentary joint-like structure when injected into a nude mouse thigh. I think that this is a nicely performed study that in principal merits publication in Nature Communications. That being said, prior to publication in Nature Communications the authors need to address the following issues:

1. The authors need to better analyze the cellular identity of the articular cartilage like cells to arise following implantation of Human synovial MSCs transduced with a Bmp7-encoding adenovirus following injection into the nude mouse thigh. The authors should employ human specific antibodies against nuclear proteins to assess whether the articular cartilage like tissue observed in these experiments is in fact of human origin. Such antibodies are commercially available (i.e., <http://www.abcam.com/human-nuclear-antigen-antibody-235-1-ab19118-1.html>; https://www.emdmillipore.com/US/en/product/Anti-Nuclei-Antibody,-clone-235-1,MM_NF-MAB1281)
2. If indeed the human synovial cells are the source of the articular-like chondrocytes, the authors should address (a) whether these cells express lubricin, (b) whether it is necessary for these cells to be transduced with a Bmp7-encoding adenovirus to generate such structures, and (c) whether implantation of MSCs from non-joint sources (similarly infected with Bmp7-encoding adenovirus) can also give rise to such joint-like structures that contain human-derived articular cartilage tissue (i.e., that expresses lubricin).

Reviewer #2 (Remarks to the Author):

Nature Communications manuscript NCOMMS-16-17866-T

This is an interesting report, with novelty and relevance, on the lineage and presence of joint morphogenetic cells in the mouse and human synovium. These cells were here shown to contribute to cartilage repair in a mouse joint injury model, and their expansion and proliferation to be responsive to the transcriptional co-factor Yap. That the adult synovium

serves as a postnatal reservoir of cells with morphogenetic ability has considerable potential clinical relevance, considering the current lack of an effective treatment that can repair the injured or osteoarthritic human joint.

The manuscript is in general focused and well written, with for the most part well supported and argued conclusions.

A few specific comments follow:

The text and figures are replete with p-values. Authors should note that statistical significance, or the level of it, does not indicate size of effect. It would be helpful to the reader to indicate, where appropriate, the magnitude of the effect or difference.

The mouse lineage experiments and the role of Yap are persuasive, while the further role of Yap needs more work considering the lack of an apparent KO-phenotype. I found it hard to discern from text or figures if mouse cartilage injury repair was impeded or not in the Yap1 cKO mice? Please clarify, and if the pivotal experiments have not been done to clarify this, their inclusion would make a good addition to the story.

The inclusion of human tissue samples enhances the study, but I note that the number of donors was small, leaving the question on individual biological- and age-related variability unanswered. Authors need to comment on this in a limitations discussion. It would be reasonable to also include an image of normal human synovium.

The final experiments on adult human MSCs to show their morphogenetic properties do not connect very well to the remainder of the manuscript: the assumption made that the cells used correspond to the mouse Gdf5 lineage is not that well supported, and the specific choice of BMP7 transduction not well argued. These experiments need to better connect to the main story, or perhaps the authors might consider removing this section and leaving it for further expansion in future report?

The discussion contains some redundancy; the writing and the 'story telling' can be tightened. Page 16 line 7, sentence starting "This points to..." can be rephrased for clarity.

I suggest some brief comments on study limitations would be appropriate in the discussion. As an example, the authors leave largely unanswered the important question of the influence of mouse strain on results (some strains are superhealers of cartilage injury). Another relevant issue is the possible influence of age of cell donor and age of individual receiving in a future cell transfer setting.

Reviewer 1

Comment 1: The authors need to better analyze the cellular identity of the articular cartilage like cells to arise following implantation of Human synovial MSCs transduced with a Bmp7-encoding adenovirus following injection into the nude mouse thigh. The authors should employ human specific antibodies against nuclear proteins to assess whether the articular cartilage like tissue observed in these experiments is in fact of human origin. If indeed the human synovial cells are the source of the articular-like chondrocytes, the authors should address (a) whether these cells express lubricin, (b) whether it is necessary for these cells to be transduced with a Bmp7-encoding adenovirus to generate such structures, and (c) whether implantation of MSCs from non-joint sources (similarly infected with Bmp7-encoding adenovirus) can also give rise to such joint-like structures that contain human-derived articular cartilage tissue (i.e., that expresses lubricin).

RESPONSE: We thank the reviewer for these very helpful comments and suggestions. To detect human cells in the implant, we performed an *in situ* hybridisation (ISH) for human-specific Alu genomic repeats. This demonstrated the presence of human nuclei in the articular cartilage and subchondral bone of the implant (see Figure 8d). To confirm these findings further and to investigate the expression of lubricin and, more in general, confirm the phenotype of the human cells in the implant, we elected to carry out RT-PCR analysis of the implant using human-specific primers. We also performed RT-PCR with mouse-specific primers to demonstrate the chimeric nature of the retrieved implant. We elected to use RT-PCR with species-specific primers for its high sensitivity and specificity that we controlled for using appropriate human-only and mouse-only controls. These new data can be found in Figure 8f. The data confirm the presence of human cells in the retrieved implant. In addition, consistent with the data obtained by ISH for human Alu repeats shown in Figure 8d, we were able to detect human lubricin, human aggrecan and human osteocalcin, indicating direct contribution of human cells to cartilage and bone, while human type X collagen (a known marker of hypertrophic chondrocytes in the growth plate) was undetectable. Detection of mouse aggrecan, mouse type X collagen and mouse osteocalcin highlights the chimeric nature of the retrieved implant. In summary, these findings indicate that the joint-like implant was of chimeric nature with contribution of human synovial MSCs to the articular surface and bone (Figure 8d).

Under similar experimental conditions, the injection of human synovial MSCs into the posterior compartment of the thigh in the nude mouse model did not result in a retrievable implant after 4 to 6 weeks. It was thus necessary for these cells to be transduced with a Bmp-encoding adenovirus to generate such structures. We have included a statement in the results to clarify this.

Finally, for cell-specificity in this proof-of-concept experiment, we used human dermal fibroblasts as mesenchymal cells from a non-joint source (De Bari et al., Arthritis Rheum 2001). We show that the injection of primary human dermal fibroblasts transduced with a Bmp7-encoding adenovirus resulted in the formation of an ossicle consisting of a perimeter of cortical bone with trabeculae and a cavity containing fatty marrow, with residual cartilage remnants (Figure 8c). Importantly, we detected few human nuclei by ISH for human Alu repeats (Figure 8e), and little or no human lubricin, aggrecan and osteocalcin (Figure 8f) indicating that the ossicle was almost entirely mouse-derived. The retrieved ossicle was histologically similar to previously reported ossicles retrieved at 4 weeks after intramuscular implantations in CD1 mice of Ad-BMP7 mixed with demineralized bovine bone powder (Franceschi et al, J Cell Biochem 2000). These findings suggest that unlike the synovial MSCs, the dermal fibroblasts made little direct contribution and failed to provide patterning signals to shape the forming structure.

Reviewer 2

Comment 1: The text and figures are replete with p-values. Authors should note that statistical significance, or the level of it, does not indicate size of effect. It would be helpful to the reader to indicate, where appropriate, the magnitude of the effect or difference.

RESPONSE: We thank the reviewer for this helpful remark. We have clarified the magnitude of the effect or difference in the text where appropriate, and p values can now be found in the Figure legends.

Comment 2: The mouse lineage experiments and the role of Yap are persuasive, while the further role of Yap needs more work considering the lack of an apparent KO-phenotype. I found it hard to discern from text or figures if mouse cartilage injury repair was impeded or not in the Yap1 cKO mice? Please clarify, and if the pivotal experiments have not been done to clarify this, their inclusion would make a good addition to the story.

RESPONSE: We have conducted an additional experiment of joint surface injury in Yap1 cKO mice. These new data can be found in Figure 6e-h. Strikingly, we detected fewer Tom⁺ cells in the repair tissue in Yap1 cKO mice compared to Yap1 cHa controls (Figure 6f). These findings demonstrate that absence of Yap1 in Gdf5 lineage cells impairs their ability to populate the cartilage injury site. We have also added a paragraph of discussion to the revised manuscript.

Comment 3: The inclusion of human tissue samples enhances the study, but I note that the number of donors was small, leaving the question on individual biological- and age-related variability unanswered. Authors need to comment on this in a limitations discussion. It would be reasonable to also include an image of normal human synovium.

RESPONSE: We have included an image of normal human synovium as recommended. In addition, we have further strengthened the data with human synovial tissues by including a triple immunofluorescence staining to demonstrate that Yap is expressed by CD55⁺ fibroblast-like synoviocytes but not CD68⁺ macrophages. Nonetheless, we appreciate the limitation of our study in relation to potential individual biological- and age-related variability that may exist. We have added a paragraph of discussion to acknowledge these limitations.

Comment 4: The final experiments on adult human MSCs to show their morphogenetic properties do not connect very well to the remainder of the manuscript: the assumption made that the cells used correspond to the mouse Gdf5 lineage is not that well supported, and the specific choice of BMP7 transduction not well argued. These experiments need to be better connected to the main story, or perhaps the authors might consider removing this section and leaving it for further expansion in future report?

RESPONSE: The finding that human synovial MSCs overexpressing Bmp7 gave rise to a joint-like structure ectopically was very surprising and exciting, and we are keen to share this with the scientific community. We have tried to integrate this finding better within the paper. In particular,

we have highlighted that culture-expanded cells from human synovium (but not dermal fibroblasts) display ability to undergo synoviogenesis in vitro (Kiener et al., Arthritis Rheum 2010), and we have confirmed this in our laboratory. We think that this unique functional feature provides a sound link between the human synovial MSCs and the mouse Gdf5-lineage synovial cells.

We used BMP7 as one of the BMP ligands known to induce endochondral ossification. BMP ligand specificity remains to be elucidated. We have added a paragraph of discussion to the revised manuscript.

Comment 5: The discussion contains some redundancy; the writing and the 'story telling' can be tightened. Page 16 line 7, sentence starting "This points to..." can be rephrased for clarity. I suggest some brief comments on study limitations would be appropriate in the discussion. As an example, the authors leave largely unanswered the important question of the influence of mouse strain on results (some strains are superhealers of cartilage injury). Another relevant issue is the possible influence of age of cell donor and age of individual receiving in a future cell transfer setting.

RESPONSE: We have restructured and tightened the discussion, and have acknowledged the limitations of the study. We have also shortened the manuscript to comply with word limits.

REVIEWERS' COMMENTS:

Reviewer #1 (Remarks to the Author):

Prior work has indicated that GDF5-Cre expressing cells in the joint can give rise to all tissues in this structure including the synovial cells, the ligaments and the articular cartilage. In the present study, the authors analyze both the proliferative and chondrogenic potential for synovial cells marked by GDF5-Cre expression. I think that this is a very carefully performed fate mapping study, which makes several interesting and novel observations: In particular, I thought that the finding that GDF5-Cre expressing synoviocytes proliferate following joint injury in a YAP-dependent fashion to be quite interesting. In addition, the authors demonstrate that GDF5-Cre expressing synoviocytes are able to reconstitute articular cartilage following joint injury in vivo, or following expansion of such cells in vitro and implantation of such cells into a joint injury model in vivo. Lastly, the authors present very striking data suggesting that Human synovial MSCs transduced with a Bmp7-encoding adenovirus can form a rudimentary joint-like structure when injected into a nude mouse thigh. I think that this revised manuscript has addressed the reviewers concerns and thus merits publication in Nature Communications in its present form.

Reviewer #2 (Remarks to the Author):

I have read the revised manuscript with pleasure. The authors have responded well to my comments and I have further requests or suggestions.
Stefan Lohmander